# Flexible Highly Thermally Conductive PCM Film Prepared by Centrifugal Electrospinning for Wearable Thermal Management

**DOI:** 10.3390/ma17204963

**Published:** 2024-10-11

**Authors:** Jiaxin Qiao, Chonglin He, Zijiao Guo, Fankai Lin, Mingyong Liu, Xianjie Liu, Yifei Liu, Zhaohui Huang, Ruiyu Mi, Xin Min

**Affiliations:** 1Engineering Research Center of Ministry of Education for Geological Carbon Storage and Low Carbon Utilization of Resources, Laboratory of Materials Utilization of Nonmetallic Minerals and Solid Wastes, National Laboratory of Mineral Materials, School of Materials Science and Technology, China University of Geosciences (Beijing), Beijing 100083, China; qiaojiaxin@jncec.com (J.Q.); hcl19980802@foxmail.com (C.H.); 15547578085@163.com (Z.G.); linfankai85@163.com (F.L.); liu188032@163.com (M.L.); liuxj1012@163.com (X.L.); lyf13016178920@163.com (Y.L.); mry0813@cugb.edu.cn (R.M.); 2Beijing Jingneng Clean Energy Co., Ltd., Beijing Branch, Beijing 100020, China

**Keywords:** phase-change materials, wearable thermal management, flexible fibrous membranes, centrifugal electrospinning, thermal conductivity

## Abstract

Personal thermal management materials integrated with phase-change materials have significant potential to satisfy human thermal comfort needs and save energy through the efficient storage and utilization of thermal energy. However, conventional organic phase-change materials in a solid state suffer from rigidity, low thermal conductivity, and leakage, making their application challenging. In this work, polyethylene glycol (PEG) was chosen as the phase-change material to provide the energy storage density, polyethylene oxide (PEO) was chosen to provide the backbone structure of the three-dimensional polymer network and cross-linked with the PEG to provide flexibility, and carbon nanotubes (CNTs) were used to improve the mechanical and thermal conductivity of the material. The thermal conductivity of the composite fiber membranes was boosted by 77.1% when CNTs were added at 4 wt%. Water-resistant modification of the composite fiber membranes was successfully performed using glutaraldehyde-saturated steam. The resulting composite fiber membranes had a reasonable range of phase transition temperatures, and the CC_4_PCF-55 membranes had melting and freezing latent heats of 66.71 J/g and 64.74 J/g, respectively. The results of this study prove that the green CC_4_PCF-55 composite fiber membranes have excellent flexibility, with good thermal energy storage capacity and thermal conductivity and, therefore, high potential in the field of flexible wearable thermal management textiles.

## 1. Introduction

With the progress of human civilization and the improvement of productivity, people are no longer worried about heat and cold and are more concerned with the styling and aesthetics of clothing, creating higher requirements for thermal comfort [1]. The emergence of indoor products such as air conditioners and electric heaters has gone a long way toward meeting the demand for thermal comfort, but higher energy consumption and greenhouse gas emissions contribute to a series of problems such as energy waste and global warming [2,3,4]. Therefore, research on wearable thermal management textiles that can effectively deal with external thermal shocks, improve the user’s thermal comfort, and store and reuse external waste heat in the form of latent heat is becoming more and more important.

In wearable thermal management textiles, phase-change materials play a leading role in storing and applying thermal energy [5]. Organic phase-change energy storage materials have been favored by more and more researchers due to their thermal stability, chemical stability, lack of phase separation, and low supercooling [6,7]. However, when used in the context of wearable thermal management, due to the low thermal conductivity of traditional organic matrices (e.g., paraffin, fatty acids, polyols) in a solid state, the resulting materials cannot be well adapted to the curvature of the human body due to the effect of rigidity, making them prone to brittle fracture and unable to meet process requirements related to flexibility [8,9]. Therefore, improving the mechanical properties (especially flexibility) and thermal conductivity of the fabricated phase-change materials is an important aspect in the study of wearable thermal management materials. At present, the preparation of flexible materials is mainly divided into internal molecular support (1. polymer blends and 2. chemical cross-linking to provide flexibility) and external backbone structure support (1. carbon-based porous materials, 2. Aerogels, or 3. phase-change fibers, among others) [10]. Hu Die et al. [11] cross-linked styrene butadiene styrene (SBS) and carbon nanotubes and filled them with paraffin wax to produce a flexible material that could be stretched and bent with a doubled elongation and maximum stress of 1.47 MPa. Wu Jiajia et al. [12] prepared PEG/PVA/CNTs phase-change films via electrostatic spinning with polyvinyl alcohol as the backbone structure and produced nanofiber films with high flexibility, breathability, and elongation of 262%. The enhancement of thermal conductivity is mainly achieved through the addition of thermally conductive particles to prepare phase-change composites. Common thermally conductive particles include alumina nanoparticles, carbon powder, carbon nanotubes, and so on. Wuri Zhang et al. [13] prepared PEG/PVP/CNTs phase-change fibers via centrifugal spinning, and the material’s thermal conductivity was greatly enhanced [to 0.265 W/(m·K)] when the carbon nanotube addition rate was 5%.

Spinning technology can blend thermally conductive fillers with polymers to obtain fibers with high aspect ratios, which is an effective way to prepare flexible thermal management materials. It is well known that electrospinning technology provides a controllable process, stable performance, and many spinning types, but the associated high voltages and long spinning times not only increase costs but also easily cause solution delamination and nanoparticle agglomeration and sedimentation, making it unconducive to large-scale production [14]. Centrifugal spinning technology is inexpensive, easy to operate, and less time-consuming, but the fiber orientation is low and the efficiency of spinning is affected by the fact that absolute homogeneity of the polymer solution cannot be guaranteed. Furthermore, the presence of air bubbles within the solution can lead to clogging of the needles [15]. Therefore, it is particularly important to select a more efficient spinning technique to prepare phase-change fibers with high enthalpy, high thermal conductivity, low leakage, non-toxicity, and good biocompatibility. Table 1 summarized the advantages and disadvantages of flexible PCM manufacturing methods.

Therefore, to solve the above problems, this study adopts the centrifugal electrospinning method, in which the polymer solution is formed into composite fiber membranes through the joint action of centrifugal and electrostatic forces. Polyethylene glycol (PEG) was selected as a phase-change material due to its non-toxicity, biocompatibility, broader phase-change temperature with higher enthalpy, and ease of chemical modification [16]. Polyethylene oxide (PEO) has good thermoplasticity, spinnability, biocompatibility, and non-toxicity, can be used as a good backbone structure, and can be cross-linked with PEG to make the composite flexible. Carbon nanotubes (CNTs) were used to enhance the mechanical properties of the phase-change fibers, as well as to increase their elongation and thermal conductivity [17]. Water-soluble polymers absorb water, swell, and disintegrate when they come into contact with water vapor. Previously reported methods to improve the water resistance of polymeric materials mainly involve chemical cross-linking, such as coating cross-linking or in situ cross-linking [18]. However, in the field of flexible wearables, coating cross-linking will block some of the inter-fiber voids and reduce the permeability of the material, while in situ cross-linking will restrict the movement of PEG chains on a large scale. In contrast, glutaraldehyde gas cross-linking modification is a mild biocompatible modification technique that improves the water resistance of the material while maintaining its flexibility and breathability. Therefore, the proposed novel green PEG/PEO/CNTs phase-change film can play an effective role in the field of wearable thermal management.

Firstly, PEG, PEO, and CNTs were blended in an aqueous solution at 70 °C to form a spinning solution, and then fibrous membranes were prepared by centrifugal electrostatic spinning. Moreover, gas cross-linking modification was carried out using glutaraldehyde to enhance its waterproof performance. An optical microscope was used to observe the morphology of the fibrous membrane, and its flexibility and strength were characterized by curling and stretching. The waterproofing performance was determined by immersing the membranes in water and testing the contact angle. Differential scanning calorimetry was performed to analyze the thermal properties of the membranes and to measure the phase transition temperature and enthalpies. Finally, the thermal management performance of the fibrous membranes was evaluated by infrared imaging and heating/cooling experiments.

## 2. Experimental Section

### 2.1. Materials

Polyethylene oxide (PEO, M_n_ = 1000), polyethylene glycol (PEG, M_n_ = 1000), glutaraldehyde (GA, 25%, *w*/*v*), and hydroxyl carbon nanotubes (with length and diameter of 10–20 μm and 10–30 nm, respectively) were purchased from Nanjing Xinfeng Nano Material Technology Co., Ltd., Nanjing, China. Distilled water was made in-house at our laboratory. None of the materials underwent further processing.

### 2.2. Preparation of Solutions for Centrifugal Electrospinning

PEO powder was dissolved into distilled water and stirred at 70 °C for 2 h to obtain a 4 wt% aqueous solution. PEO/PEG solutions were obtained by adding PEG at different PEO/PEG mass ratios of 100/0, 60/40, 55/45, 50/50, 45/55, and 40/60. In addition, the PEO/PEG/CNTs spinning solutions were prepared with a fixed PEO/PEG mass ratio of 45/55, and the concentration of CNTs was increased linearly (0, 2, 4, 6, 8, and 10 wt%) with respect to the PEO content.

### 2.3. Preparation of Fiber Membrane by Centrifugal Electrospinning

The composite fiber membrane was prepared and assembled by centrifugal electrospinning in our laboratory [15]. We added an electrostatic field to alleviate the rapid spinning of well-formed fibers. The spinning needles were 27GTT oblique needles, and the speed of the motor was 2500 rpm. The applied voltage was 15 kV. During the spinning process, the ambient temperature was controlled at 60 °C, and the relative humidity was controlled at 30 ± 5%. The spinning solution was put into the spinneret hole through a syringe and was squeezed and sprayed out to form fibers at a high rotation speed. The composite fibrous membranes were obtained under an electric field. The obtained composite fibrous membranes with different PEO/PEG mass ratios are denoted as PCF-y, where y represents the mass ratio of PEG. Furthermore, the PEO/PEG/CNTs composite fibrous membranes with different ratios of CNTs added are denoted by C_x_PCF-y, where x denotes the mass ratio of CNTs relative to PEO.

### 2.4. Gas Cross-Linking Modification of Glutaraldehyde

For this experiment, glutaraldehyde solution was used to modify the composite fibrous membranes for water resistance. A certain amount of glutaraldehyde solution was added to a desiccant tank, and the composite fibrous membranes were placed in the middle of the desiccant tank with a screen mesh, sealed, and moved into an oven at 40 °C for 24 h. Due to the volatility of glutaraldehyde, the hydrophilic groups of PEO and PEG reacted with glutaraldehyde-saturated vapor in a condensation reaction to complete the water resistance modification of the composite fibrous membranes. The modified membranes were dried under vacuum at room temperature for 24 h. The composite fibrous membranes modified by glutaraldehyde are denoted as CC_x_PCF-y.

### 2.5. Characterization

The surface morphology and microstructure of the composite fibrous membranes were determined using an optical microscope. The chemical structure of the composite fibrous membranes was studied with a Fourier-transform infrared spectrometer (FT-IR, Spectrum 1, PerkinElmer, Waltham, MA, USA) in the wavelength range of 400–4000 cm^−1^. The crystalline properties of the composite fibrous membranes were analyzed through X-ray diffraction (XRD, D8 Advance, Bruker, Karlsruhe, Germany) in the angular range of 5–90°. The optical properties of the composite fibrous membranes were analyzed using a UV–Vis–NIR spectrometer (UV–Vis–NIR, UV-3600Plus, Shimadzu, Kyoto, Japan). Modification of the hydrophilic properties of the composite fibrous membranes before and after glutaraldehyde modification was evaluated using a video contact-angle analyzer. The composite fibrous membranes were heated with pure PEG on a hot plate at 50 °C to analyze their shape stability. The phase transition temperature and enthalpy of phase transition of the composite fibrous membranes were measured using a differential scanning calorimeter (DSC Q2000, TA, Newcastle, DE, USA) in a nitrogen atmosphere with a flow rate of 50 mL/min. The membranes were heated or cooled in the range of −10 to 70 °C, with a temperature change rate of 5 °C/min. A multi-channel data recorder (TP720, Tuopu, Tianjin, China) and an infrared thermal imaging camera (FLIR ONE, FLIR, Wilsonville, OR, USA) recorded the temperature evolution of the composite fibrous membranes during warming and cooling. The thermal conductivities of the composite fibrous membranes were tested using a laser thermal conductivity analyzer (LINSEIS XFA 500, Linseis, Selb, Germany).

## 3. Results and Discussion

In this paper, PEG and PEO were blended and cross-linked, and CNTs were added to enhance their thermal conductivity, following which composite fiber membranes with excellent flexibility, thermal properties, anti-leakage properties, shape stability, and favorable thermal management performance were prepared by centrifugal electrospinning. Finally, the composite fibrous membranes were modified by gas cross-linking using GA to improve their waterproof performance. In this section, the morphology, flexibility, mechanical strength, water resistance, thermal conductivity, leakage resistance, thermal performance, thermal cycling stability, and thermal management performance of the as-prepared composite fiber membranes are discussed, confirming their promising potential in practical applications.

### 3.1. Preparation Strategy of Composite Fiber Membranes

The composite fibrous membranes were designed according to the following principles: (i) the spinning process should have high efficiency and low cost, without using harmful solvents [19]; (ii) the fibrous membrane should possess high flexibility, in order to adapt to the body’s curvature, and high breathability [20]; and (iii) the composite fiber membrane should possess a suitable phase-change temperature and high phase-change enthalpy, enhancing its potential for practical application. In this study, the PEO/PEG spinning solution was prepared into composite fibrous membranes using environmentally friendly and efficient centrifugal electrospinning technology, as shown in Figure 1. Interconnected porous structures were formed in the composite fiber membranes. Most importantly, hydrogen bonding and glutaraldehyde modification endowed the composite fibrous membranes with a stable chemical structure and water resistance.

### 3.2. Morphology and Structure Analysis of Composite Fibrous Membranes

The surface morphology of the composite fibrous membranes and single fibers with different mass ratios of PEG and CNTs was analyzed using an optical microscope. Figure 2a–l show that the composite fibrous membranes presented a porous structure with randomly oriented fibers, where the interconnected pore structure enhanced the breathability of the membranes. Meanwhile, the pristine PEO membranes were loosely structured, and the fibers were not stretched sufficiently. With an increase in PEG content, the average diameter of the composite fiber gradually increased, and smooth, uniform, and cylindrical fibers were formed, which indicated that PEG and PEO have good compatibility and co-spinnability. The results show that, at a PEG content of 60 wt%, the composite fibers adhered to each other and the composite fibrous membranes had poor morphology. Considering that the composite fibrous membranes should contain as much PEG as possible, in order to enhance the thermal storage performance of the membranes [21], 55 wt% was chosen as the optimal PEG content. With an increase in the CNTs content, the viscosity of the spinning solution and the diameter of the fibers increased. When the content of CNTs increased, an agglomeration phenomenon was observed due to their poor dispersion in the spinning solution, which affected the surface morphology of the composite fibrous membranes as partially agglomerated CNTs blocked the interconnected pore structure. The fibrous and porous morphology of CC_4_PCF-55 was still well preserved after modification with glutaraldehyde. The process of vapor-phase modification did not cause significant damage to the composite fibrous membranes, and the 3D polymer network was well preserved.

### 3.3. Mechanical Properties of Composite Fibrous Membranes

Mechanical properties are a key factor in the practical application of materials. Composite fiber membranes will inevitably be folded, curled, and stretched in the course of their use [22]. As shown in Figure 3a–c, the composite fiber membranes remained intact after folding and curling. One gram of the composite fiber membrane could successfully pull up to 500 g of weight. The excellent flexibility and toughness of the composite fiber membranes demonstrate that they can effectively adapt to the curvature changes of human skin while withstanding stretching and twisting in the actual application process, indicating their great potential in the field of flexible wearable thermal management textiles.

### 3.4. Water Resistance Properties of Composite Fibrous Membranes

In this experiment, glutaraldehyde was introduced to modify the composite fibrous membranes for water resistance. The hydroxyl groups at the ends of PEG and PEO can react with the aldehyde group in glutaraldehyde in a hydroxyl–aldol condensation reaction. This reduces the number of hydrophilic groups in the composite fibrous membranes, thus preventing them from absorbing water and swelling and disintegrating after contact with water or water vapor [23,24]. As presented in Figure 3d–f, the composite fibrous membranes retained their shape stability after 5 min and 10 min of immersion in water, and no obvious structural damage occurred.

FT-IR spectroscopy (Figure 3g) confirmed the occurrence of an acetal reaction during modification of the composite fibrous membranes. The broad characteristic absorption peaks centered at 3438 cm^−1^ are the contraction vibration peaks of —OH groups, the stretching vibration peaks of the —CH_2_ groups are at 2884 cm^−1^, and the strong characteristic peaks near 1105 cm^−1^ are the C—O—C stretching vibration peaks. As the FT-IR spectrum shows, after modification with glutaraldehyde, the characteristic absorption peaks of —OH (3438 cm^−1^) were significantly weakened, while the characteristic absorption peaks of C—O—C (1105 cm^−1^) were significantly stronger. The change in the characteristic absorption peaks confirmed that the acetal reaction was successfully completed. As shown in Figure 3h,i, the contact angle of the composite fibrous membranes increased from 40.0° to 70.8° after the modification. This increase in contact angle confirms the transition from water solubility to hydrostability of the composite fibrous membranes, further confirming that glutaraldehyde was successful in modifying the composite fibrous membranes to promote their water resistance.

### 3.5. Thermal Conductivity of Composite Fibrous Membranes

The thermal storage efficiency of composite fibrous membranes is affected by their thermal conductivity [25]. High thermal conductivity enables the composite fibrous membranes to respond to changes in the external environment in a timely manner. As shown in Figure 4a, the thermal conductivity of PCF-55 was 0.0354 W/(m·K). Compared with PCF-55, the thermal conductivity of C_10_PCF-55 was increased by 154.8%. Furthermore, the thermal conductivity of C_4_PCF-55 improved by 73.2% when CNTs were added at 4 wt%, demonstrating that the thermal conductivity of the composite fibrous membranes increased regularly with increasing CNTs content. CNTs can be used as a good heat-conducting filler to provide heat transfer channels after co-blending with a spinning solution [26]. The highly thermally conductive CNTs enhanced the thermal conductivity of the composite fibrous membranes through the creation of a connected, thermally conductive network. The thermal conductivity of the glutaraldehyde-modified CC_4_PCF-55 was 0.0627 W/(m⋅K), and the thermal conductivity was improved by 77.1%, similarly to that for C_4_PCF-55. The similarity of the thermal conductivity further confirmed that the mild gas modification technique does not damage the three-dimensional mesh structure of the composite fibrous membranes.

### 3.6. Thermal Stability and Crystalline Properties of Composite Fibrous Membranes

The crystalline properties of PEG and PEG-embedded composite fibrous membranes were studied using the X-ray diffraction technique. Figure 4b presents the XRD patterns of PEG, C_4_PCF-55, and CC_4_PCF-55. The two diffraction peaks at 19.2° and 23.3° for PEG correspond to the lattice planes (120) and (132), respectively. The higher diffraction peaks indicate that PEG has good crystalline properties. The diffraction pattern of C_4_PCF-55 has similar diffraction peaks to PEG, at around 19.2° and 23.3°, confirming that PEG can still act as a functional component after embedding, thus ensuring the phase-change heat storage capacity of the composite fibrous membranes. The composite fibrous membranes presented good crystalline properties: compared with C_4_PCF-55, the diffraction peak position for CC_4_PCF-55 did not change, and no new diffraction peaks appeared. This indicates that the gas modification technique did not damage the lattice of the composite fibrous membranes. On the other hand, the decrease in the intensity of the diffraction peak near 19.2° indicated that the crystalline properties of the composite fibrous membranes were reduced due to the acetal reaction, which limited the free movement of the PEG fragments, thus sacrificing some of the heat storage capacity to enhance the thermal stability of the composite fibrous membranes [27].

Preventing the leakage of PEG from composite fibrous membranes is important to obtain a continuous and stable heat storage capacity, and it is also a key factor in increasing the practical applicability of composite fibrous membranes. PEG and CC_4_PCF-55 were co-located on a heating plate at 50 °C for leakage experiments, and the shape stability and thermal stability of the composite fibrous membranes were determined by comparing their shape changes. As shown in Figure 4c, with the passage of time, the PEG gradually changed from a white lumpy solid to a transparent one and, finally, to liquid PEG after 6 min. During the heating process, CC_4_PCF-55 maintained its original shape, and no leakage of PEG embedded in the fibrous membranes was observed. This is because the PEG chains were physically cross-linked with the PEO backbone structure and, together, they formed a 3D polymer network under the hydrogen bonding and acetal reactions, preventing the leakage of PEG through physical entanglement and chemical bonding. This indicates that the composite fibrous membranes had a certain degree of thermal and shape stability, and that they had better performance in terms of thermal energy storage and recycling. Therefore, the prepared composite fibrous membranes have high potential for application in the field of flexible wearable thermal management textiles.

### 3.7. Phase Transition Temperature and Thermal Storage Capacity of Composite Fibrous Membranes

The phase-change temperature and heat storage capacity are key factors affecting the application of heat storage materials [28]. The phase transition temperature and heat storage capacity of the composite fiber membranes were measured through the DSC test. The melting/freezing latent heats (ΔH_m_/ΔH_f_) and melting/freezing temperatures (T_m_/T_f_) of the composite fiber membranes are presented in Figure 5a,b. Endothermic/exothermic peaks for PEG and the various types of composite fiber membranes were observed during warming and cooling, and the process was reversible. This is mainly attributable to the transformation of the two crystal forms: the endothermic peak corresponds to solid–liquid phase transitions, while the exothermic peak corresponds to liquid–solid phase transitions. The melting/freezing latent heats (ΔH_m_/ΔH_f_) of pure PEG were measured to be 145.9 J/g and 142.4 J/g in the DSC test, respectively, and the melting/freezing temperatures (T_m_/T_f_) were 42.8 °C and 27.3 °C, respectively. The melting/freezing latent heats correspond to the heat storage capacity of the composite fiber membranes, which reflects the two-way heat regulation function, while the melting/freezing temperatures affect the application scenarios of the composite fiber membranes.

The composite fiber membranes exhibited good phase-change enthalpy with suitable phase-change temperatures. Furthermore, their melting/freezing temperatures were within the range of human comfort, meaning that these composite fiber membranes have good potential for application in the context of personal thermal management. The low latent heats of melting/freezing of the composite fiber membranes, compared to those of pure PEG, are in part due to the action of PEO in the 3D polymer network and in part due to the CNTs replacing the PEG as a thermally conductive filler, thus reducing the mass ratio of PEG in the composite fiber membranes.

The latent heats of melting/freezing for the composite fiber membranes increased with the addition of CNTs and peaked at 4% CNTs. This was attributed to the fact that the CNTs were dispersed in the polymer network as thermally conductive fillers, which provided sites for heterogeneous nucleation and promoted the crystallization process of PEG, thus increasing the enthalpy of phase transition in the composite fiber membranes. As the content of CNTs continued to increase, the latent heat of melting/freezing of the composite fiber membranes gradually decreased due to the constraining effect of excess CNTs on the PEG chains, hindering further growth of the PEG crystals [29]. In practice, supercooling is a key parameter, which is the difference between the melting and crystallization temperatures of a material [30].

As shown in Figure 5c, the melt-phase transition temperature of the composite fiber membranes decreased and their degree of supercooling also reduced. This was due to the chemical cross-linking of PEG with PEO to promote the reduction in its melting phase transition temperature, and the addition of CNTs acted not only as a heterogeneous nucleating agent but also as a heat transfer medium to improve the thermal conductivity and promote heat transfer [31]. The enthalpies of melting and crystallization of C_4_PCF-55 were 69.37 J/g and 68.65 J/g, respectively; those of CC_4_PCF-55 were 66.71 J/g and 64.74 J/g, respectively. Compared with the calculated theoretical values (80.25 J/g and 78.32 J/g), they decreased by 13.6%, 12.3%, 16.9%, and 17.3%, respectively. This suggests that a small portion of the PEG did not participate in the phase transition process during heating and freezing [32]. This was primarily attributed to the hydrogen bonding effect and acetal reaction between PEG and PEO, which resulted in a small portion of PEG chains existing in an amorphous state. Moreover, the solvent evaporated rapidly during centrifugal electrospinning, and the PEG chains could not form a well-arranged crystal in the 3D polymer network. The cross-linking of GA with PEO and PEG further limited the movement of PEG and hindered the formation of crystalline reactions, causing a further reduction in the enthalpy.

Table 2 presents the comparison of the experimentally produced composite fiber membranes with previously reported fiber membranes in terms of precursor solvent, phase transition temperature, and phase transition enthalpy. There have been several studies on phase-change thermal storage fibers. Most of the previous electrostatic spinning techniques have been carried out using organic solvents, and few works have investigated green composite fiber membranes. Typically, more than 80% of the organic solvent is consumed in the PCF preparation process. The organic substances used as solvents (e.g., DMF, DMAC, formic acid, and acetone) are evaporated during the spinning process and emitted directly into the air. Notably, these organic solvents are costly and are associated with environmental concerns.

As a result, green composite fiber membranes were prepared in one step through efficient centrifugal electrostatic spinning, using distilled water as the solvent. The resulting composite fiber membranes were found to have comparable latent heat to those using organic solvents, as well as being less costly. It is worth mentioning that the phase-change temperature range of CC_4_PCF-55 composite fiber membranes is 27.5–33.9 °C, which is within the human comfort temperature range. Composite fiber membranes with large heat storage capacity, with a suitable temperature range, and formed through a non-toxic, green, and efficient preparation method have high potential for application in the context of personal thermal management.

### 3.8. The Thermoregulatory Capacity of the Composite Fiber Membranes

The experimentally prepared CC_4_PCF-55 composite fiber membranes exhibited excellent temperature regulation capacity. An IR camera and a multiplexed data logger were used to record the temperature versus time course of the CC_4_PCF-55 composite fiber membranes, which were compared with pure PEO membranes. As shown in Figure 6, the color of the PEO membranes shifted rapidly from blue to red during the heating process, reaching a peak after 400 s of heating. The CC_4_PCF-55 composite fiber membranes slowed down the warming at the phase transition temperature and gradually reached the peak after 500 s. This means that the composite fiber membranes absorbed more heat than the PEO membranes. A similar delayed heat was observed in the fiber membranes during the cooling phase. The composite fiber membranes showed a significant plateau at 350 s, and the time to cool to room temperature was increased by 80% compared to the PEO membranes. The increasing and decreasing temperature curves further confirm that the composite fiber membranes have better temperature regulation capability and, thus, greater potential in the fields of personal thermal management and electronic component insulation.

## 4. Conclusions

In this work, a green PEG/PEO/CNTs composite fiber phase-change membrane was fabricated by centrifugal electrospinning, where PEG acted as the phase-change material to provide the energy storage density, while PEO provided the three-dimensional backbone to load the PEG and cross-linked with the PEG to form excellent flexibility, CNTs improved its thermal conductivity, and the material was modified with GA to improve its water repellency. The as-prepared CC4PCF-55 composite fiber membrane exhibited favorable energy density, with an enthalpy of 66.71 J/g, excellent thermal cycling stability, and anti-leakage properties. It possessed a thermal conductivity of 0.0627 W/(m·K), which was 77.1% higher than that of the PCF-55 sample without added CNTs. Meanwhile, the composite phase-change membrane demonstrated superior thermal management and temperature regulation capability, suggesting that this material has great potential for application in the fields of smart wearables and thermal management of electronic devices.

## Figures and Tables

**Figure 1 materials-17-04963-f001:**
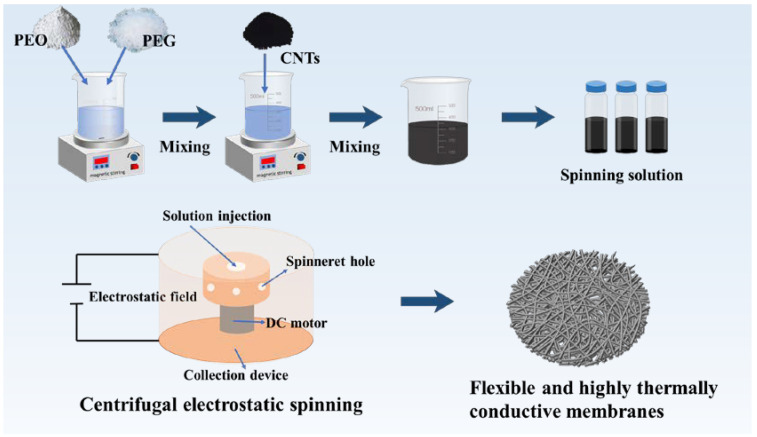
Centrifugal electrostatic spinning for the preparation of flexible and highly thermally conductive phase-change thermal storage membranes.

**Figure 2 materials-17-04963-f002:**
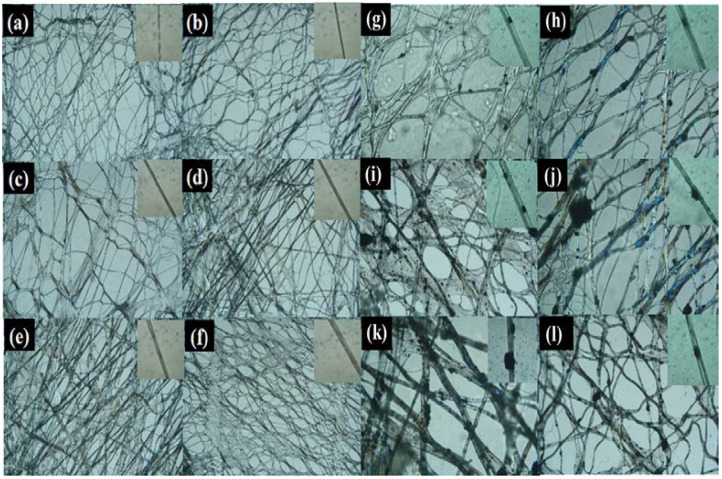
Images of (**a**) PEO, (**b**) PCF-40, (**c**) PCF-45, (**d**) PCF-50, (**e**) PCF-55, (**f**) PCF-60, (**g**) C_2_PCF-55, (**h**) C_4_PCF-55, (**i**) C_6_PCF-55, (**j**) C_8_PCF-55, (**k**) C_10_PCF-55, and (**l**) CC_4_PCF-55 fibrous membranes and the corresponding single fiber.

**Figure 3 materials-17-04963-f003:**
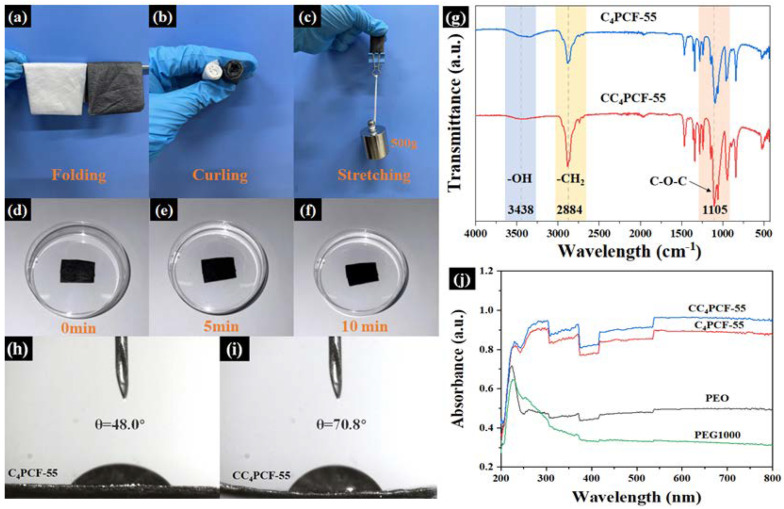
(**a**–**c**) Folding, curling, and stretching images, respectively, of C_4_PCF-55 and CC_4_PCF-55 fibrous membranes. (**d**–**f**) Optical images of a CC_4_PCF-55 membrane immersed in water for 0 min, 5 min, and 10 min, respectively. (**g**) FT-IR spectra of C_4_PCF-55 and CC_4_PCF-55 fibrous membranes. (**h**,**i**) Water contact angle images of C_4_PCF-55 and CC_4_PCF-55 fibrous membranes, respectively. (**j**) UV—visible spectrogram of C_4_PCF-55 and CC_4_PCF-55 fibrous membranes.

**Figure 4 materials-17-04963-f004:**
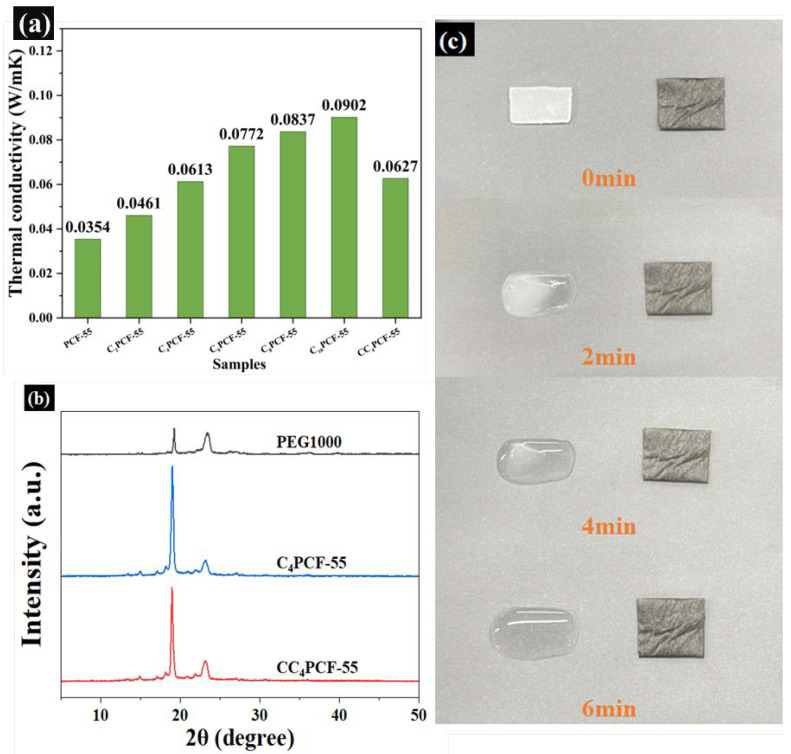
(**a**) Thermal conductivity of the composite fibrous membranes. (**b**) XRD patterns of PEG1000, C_4_PCF-55, and CC_4_PCF-55. (**c**) Shape stability of PEG1000 and CC_4_PCF-55.

**Figure 5 materials-17-04963-f005:**
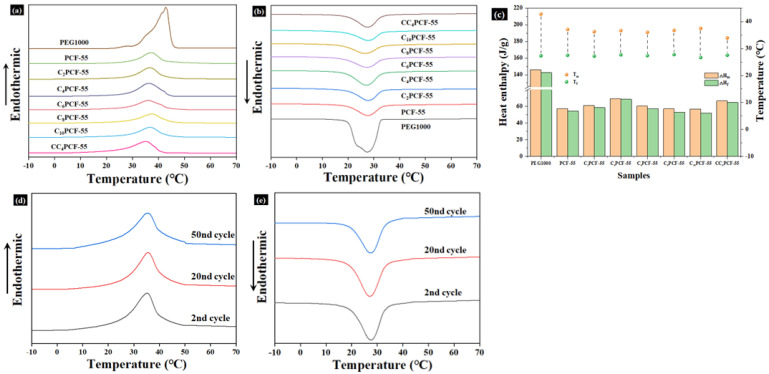
Thermal properties of the prepared composite fiber membranes: DSC curves of PEG and the composite fiber membranes for (**a**) the endothermic process and (**b**) the exothermic process. (**c**) Melting/freezing enthalpies (ΔH_m_/ΔH_f_) and melting/freezing phase transition temperatures (T_m_/T_f_) of the composite fiber membranes. DSC curves of the CC_4_PCF-55 fiber membrane after 2, 20, and 50 cycles in the (**d**) endothermic process and (**e**) exothermic process.

**Figure 6 materials-17-04963-f006:**
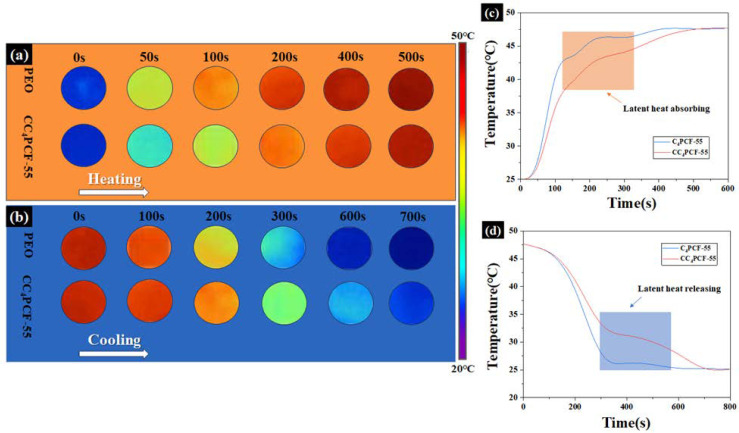
Representative thermal images of the PEO and CC_4_PCF55 membranes recorded by an IR camera in the (**a**) heating and (**b**) cooling processes. (**c**,**d**) Temperature evolution graphs of the PEO and CC_4_PCF55 fiber membranes.

**Table 1 materials-17-04963-t001:** Advantages and disadvantages of methods for fabricating flexible PCMs.

Manufacturing Method	The Advantages and Disadvantages
Polymer blends	Advantages: high energy storage density, good mechanical strength, easy to prepare Disadvantages: low thermal conductivity, loss of flexibility at temperatures below the phase transition temperature
Chemical cross-linking	Advantages: high energy storage density, excellent mechanical tensile strength, good chemical stability, favorable shape stability Disadvantages: low thermal conductivity, reduced thermal performance
Aerogel	Advantages: high thermal conductivity, multi-response function, great stability and durability Disadvantages: high cost and energy consumption, relatively poor mechanical strength
Carbon-based porous materials	Advantages: shape-memory properties, well-controlled customisability Disadvantages: easy leakage, low thermal conductivity
Electrospinning	Advantages: controllable dimensions, suitable for wearability, enables production of nanoscale fibres Disadvantages: poor energy storage capacity, low thermal conductivity, low production efficiency
Centrifugal spinning	Advantages: controllable dimensions, suitable for wearability, high production efficiency Disadvantages: poor energy storage capacity, low thermal conductivity, spinning instability
Centrifugal electrospinning	Combining the advantages of centrifugal spinning and electrostatic spinning

**Table 2 materials-17-04963-t002:** Melting enthalpy and temperature of previous phase-change fibers.

PCFs	Solvent	T_m_ (°C)	ΔH_m_ (J/g)	Reference
PEG1500/CA	DMAc/acetone	44.2	39.5	[33]
PEG600/PEG1000/PVDF	DMAc/acetone	29.3	48.9	[34]
PEG1000/PVDF/SiO_2_	DMAc/acetone	38.2	59.2	[35]
PEG10000/PHBV	DMF	57.6	62.8	[36]
PEG4000/PVDF	DMF	62.8	68	[37]
PEG6000/PU	DMF/THF	53.9	60.4	[38]
PA6/PEG1000/TiO_2_	Formic acid	36.6	51.1	[39]
PVP/PEG/Al_2_O_3_	Ethanol	47.4	54.3	[40]
PEG1000/PVA/CNTs	Water	38.9	60.1	[12]
C_4_PCF-55	Water	36.24	69.4	This work
CC_4_PCF-55	Water	33.9	66.7	This work

## Data Availability

The data presented in this study are available on request the corresponding author.

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
