# Peer review of "Flexible Highly Thermally Conductive PCM Film Prepared by Centrifugal Electrospinning for Wearable Thermal Management"

_materials, 2024, doi:10.3390/ma17204963_

Round 1

Reviewer 1 Report

Comments and Suggestions for Authors

1. The title should clearly and concisely communicate the main focus and contribution of your research. Consider incorporating keywords that accurately reflect the paper's content and potential impact. A strong title will attract readers and effectively summarize the paper's purpose.

2. To introduce the abbreviation "CNTs" and ensure clarity, please use "carbon nanotubes (CNTs)" on its first appearance in the abstract.

3. To enhance the discoverability of your paper, consider expanding the keyword list to include additional terms relevant to your research.

4. Adding a table of abbreviations between the abstract and the introduction would enhance the document. Due to the prevalence of numerous abbreviations, add that. This will help readers quickly reference and understand the meanings of these abbreviations as they read through the document. 

5. To enhance readability of the literature review, consider including a table summarizing previous works. This table could be organized chronologically to provide a clear progression of research in this area.

6. To provide readers with a clear roadmap of the paper, consider adding a brief overview paragraph at the end of Section 1. This paragraph should outline the structure of the subsequent sections, highlighting the key topics covered in each.

7. To enhance visual understanding and engagement, consider incorporating relevant photos or illustrations into Section 2. These visual aids can effectively complement the textual content and help readers grasp complex concepts more easily. Additionally, section 2 would benefit from additional references to support the presented information and establish a stronger foundation for the subsequent discussion. Please cite relevant references.

8. Figure 4 is currently difficult to read. To improve clarity, please consider replotting.

9. In the conclusion, you give a detailed summary of your current research findings. In order to organize them into a more structured framework, it might be helpful to suggest additional perspectives for future work and categorize them according to their importance or urgency for the field.

10. To enhance the paper's structure and readability, consider adding a brief introductory paragraph at the beginning of Section 3. This paragraph should outline the main themes or objectives of the section and how they connect to the overall research.

11. Please identify the instrument located in the lower left corner of Figure 1 by its specific name or type.

12. in the introduction, it is better to first have a quick review to other thermal storage modification methods such as: Numerical investigation of rectangular thermal energy storage units with multiple phase change materials; Thermodynamic analysis of heat storage of ocean thermal energy conversion integrated with a two-stage turbine by thermal power plant condenser output water, to compare it with your method. There are some other previous related works in literature, which are good to read.

Comments on the Quality of English Language

Please read carefully the text and search for typos ("a" "," "the", Capital letter, space between word.)  

Author Response

Comments from the Reviewer 1:

  1. The title should clearly and concisely communicate the main focus and contribution of your research. Consider incorporating keywords that accurately reflect the paper's content and potential impact. A strong title will attract readers and effectively summarize the paper's purpose.

Reply: Thank you very much for your valuable comments, we have simplified and highlighted the title. The changed title is “Flexible Highly Thermally Conductive PCM Film by Centrifugal Electrospinning for Wearable Thermal Management”.

  1. To introduce the abbreviation "CNTs" and ensure clarity, please use "carbon nanotubes (CNTs)" on its first appearance in the abstract.

Reply: We appreciate the reviewer's advice. Many thanks to the reviewer for the reminder and we have included the full name of CNTs in the abstract.

In this work, polyethylene glycol (PEG) was chosen as the phase change material, polyethylene oxide (PEO) was used as the backbone structure of the 3D polymer network, and carbon nanotubes (CNTs) were used to improve the mechanical properties and thermal conductivity of the material.

  1. To enhance the discoverability of your paper, consider expanding the keyword list to include additional terms relevant to your research.

Reply: We appreciate the reviewer's advice. We've made changes to the keywords.

Keywords: Phase change materials; Wearable thermal management; Flexible fibrous membranes; Centrifugal electrospinning; Thermal conductivity

  1. Adding a table of abbreviations between the abstract and the introduction would enhance the document. Due to the prevalence of numerous abbreviations, add that. This will help readers quickly reference and understand the meanings of these abbreviations as they read through the document.

Reply: Thank you for the suggestion. We think the reviewer's suggestion is very valid, so we have added a table of abbreviations in the document.

Nomenclature

Full names

Abbreviations

Full names

Abbreviations

phase change materials

PCMs

PEO/PEG/CNTs composite fibrous membranes with different ratios of CNTs added

CxPCF-y

carbon nanotubes

CNTs

the composite fibrous membranes modified by glutaraldehyde

CCxPCF-y

polyethylene glycol

PEG

Fourier transform infrared spectrometer

FT-IR

polyethylene oxide

PEO

X-ray diffraction

XRD

glutaraldehyde

GA

UV-Vis-NIR spectrometer

UV-Vis-NIR

different PEO/PEG mass ratios

PCF-y

differential scanning calorimeter

DSC

  1. To enhance readability of the literature review, consider including a table summarizing previous works. This table could be organized chronologically to provide a clear progression of research in this area.

Reply: Thank you so much for your advice.

Table1 Advantages and disadvantages of methods for fabricating flexible PCMs

Manufacturing method

The advantages and disadvantages

Polymer blends

Advantages: high energy storage density, well mechanical strength, easy to prepare

Disadvantages: low thermal conductivity, loss of flexibility at temperatures below the phase transition

Chemical cross-linking

Advantages: high energy storage density, excellent mechanical tensile strength, well chemical stability, favourable shape stability

Disadvantages: low thermal conductivity, Reduced thermal performance

Aerogel

Advantages: high thermal conductivity, multi-response function, great stability and durability

Disadvantages: high cost and energy consumption, relatively poor mechanical strength

Carbon-based porous materials

Advantages: shape memory property, well-controlled customisability

Disadvantages: leakage easily, low thermal conductivity

Electrospinning

Advantages: controllable dimensions, suitable for wearability, enables production of nanoscale fibres

Disadvantages: poor energy storage capacity, low thermal conductivity, low production efficiency

Centrifugal spinning

Advantages: controllable dimensions, suitable for wearability, high production efficiency

Disadvantages: poor energy storage capacity, low thermal conductivity, spinning instability

Centrifugal electrospinning

Combining the advantages of centrifugal spinning and electrostatic spinning

  1. To provide readers with a clear roadmap of the paper, consider adding a brief overview paragraph at the end of Section 1. This paragraph should outline the structure of the subsequent sections, highlighting the key topics covered in each.

Reply: We are very grateful for your comments. In response to your comments, we are adding a brief overview paragraph at the end of section 1. It is listed here:

Firstly, PEG, PEO and CNTs were blended in an aqueous solution at 70°C to form a spinning solution, and then fibrous membranes were prepared by centrifugal electrostatic spinning. Moreover, gas cross-linking modification was carried out by glutaraldehyde to enhance its waterproof performance. Optical microscope was used to observe the morphology of the fibrous membrane, and its flexibility and strength were characterised by curling and stretching. Waterproofing performance was determined by immersing the membranes in water and testing the contact angle. Differential scanning calorimetry was performed to analyse the thermal properties of the membranes and to measure the phase transition temperature and enthalpies. Finally, the thermal management performance of the fibrous membranes was evaluated by infrared imaging and heating/cooling experiments.

  1. To enhance visual understanding and engagement, consider incorporating relevant photos or illustrations into Section 2. These visual aids can effectively complement the textual content and help readers grasp complex concepts more easily. Additionally, section 2 would benefit from additional references to support the presented information and establish a stronger foundation for the subsequent discussion. Please cite relevant references.

Reply: We are grateful for the reviewer’s remarks. To make it easier to understand the preparation process of fibrous membranes, a schematic diagram of the centrifugal spinning technique from previous work on the subject is cited in the article. We have added an electrostatic field to improve the centrifugal spinning system, which made it easier and faster to spin well-formed fibres.

  1. Figure 4 is currently difficult to read. To improve clarity, please consider replotting.

Reply: Appreciate your advice very much! We have reworked Figure 4.

Figure 5. (a) Thermal conductivity of the composite fibrous membranes. (b) XRD patterns of PEG1000, C4PCF-55, and CC4PCF-55. (c) Shape stability of PEG1000 and CC4PCF-55.

  1. In the conclusion, you give a detailed summary of your current research findings. In order to organize them into a more structured framework, it might be helpful to suggest additional perspectives for future work and categorize them according to their importance or urgency for the field.

Reply: We are grateful to the reviewer for the wise suggestion. We have rewritten the conclusion section of this paper to facilitate a clearer understanding of the research findings in this paper. It is listed here:

In this work, a green PEG/PEO/CNTs composite fibre phase change membrane was fabricated by centrifugal electrospinning method, in which PEO and PEG were crosslinked with each other to form excellent flexibility and to prevent PEG from leaking, CNTs improved its thermal conductivity, and modified with GA to improve its water repellency. The as-prepared CC4PCF-55 composite fibre membrane exhibited favourable energy density with an enthalpy of 66.71 J/g, excellent thermal cycling stability and anti leakage. It possessed a thermal conductivity of 0.0627 W/ (m·K), which was 77.1% higher than that of the PCF-55 sample without added CNTs. Meanwhile, the composite phase change membrane demonstrated superior thermal management and temperature regulation capability, suggesting that it has great potential for application in the fields of smart wearable and thermal management of electronic devices.

  1. To enhance the paper's structure and readability, consider adding a brief introductory paragraph at the beginning of Section 3. This paragraph should outline the main themes or objectives of the section and how they connect to the overall research.

Reply: We agree with the reviewer's opinion. Following your comments, we have added a brief introductory paragraph at the beginning of Section 3, and it is listed here.

In this paper, PEG and PEO were blended and crosslinked, and CNTs were added to enhance their thermal conductivity, following which composite fibre membranes with excellent flexibility, thermal properties, anti-leakage, shape stability and favorable thermal management performance were prepared by centrifugal electrospinning method. Finally, the composite fibrous membranes were modified by gas cross-linking using GA to improve their waterproof performance. In this section, the morphology, flexibility and mechanical strength, water resistance, thermal conductivity, leakage resistance, thermal performance, thermal cycling stability and thermal management performance of the as-prepared composite fibre membranes were respectively measured, which confirmed their promising potentials in practical applications.

  1. Please identify the instrument located in the lower left corner of Figure 1 by its specific name or type.

Reply: Thank you very much for your question. The lower left instrument in Figure 1 is a simulation of a centrifugal spinning machine, and We have labelled the name in the figure.

  1. In the introduction, it is better to first have a quick review to other thermal storage modification methods such as: Numerical investigation of rectangular thermal energy storage units with multiple phase change materials; Thermodynamic analysis of heat storage of ocean thermal energy conversion integrated with a two-stage turbine by thermal power plant condenser output water, to compare it with your method. There are some other previous related works in literature, which are good to read.

Reply: Thank you very much for your comments. The literature ‘Numerical investigation of rectangular thermal energy storage units with multiple phase change materials’ was the utilization of warm water output from the condenser of a thermal power plant to replace surface water in order to improve the OTEC cycle output power, thermal efficiency, heat storage etc. The literature ‘Thermodynamic analysis of heat storage of ocean thermal energy conversion integrated with a two-stage turbine by thermal power plant condenser output water’ was to study the effect of the shape and size etc. of the PCMs on the heat transfer efficiency. However, this paper is about the preparation of composite fibre phase change membranes with excellent flexibility and phase change enthalpy for broadening the application scenarios of PCMs in areas such as personal thermal management. Thus the relevant studies in the above two literatures are not consistent with the main purpose of this paper and are not presented in the introduction. We hope to have your agreement.

Reviewer 2 Report

Comments and Suggestions for Authors

This manuscript introduces a composite fiber membrane for PCM applications, which is a crucial topic in thermos engineering. While the topic is interesting, its value is diminished by the presence of unrefined English, numerous typos, and grammatical mistakes. For example,

In Page 3: can not (àcannot)

Introduction: When citing reference, write only family name of first author.

In page 8: Figure 2a-1

In Page 9: ~. 1g of ~ “In English writing, a sentence does not begin with number. (à One g of~)

In Page 11: “the change in the characteristic absorption peaks confirmed the acetal reaction was suceesfully conducted”

As shown in Figure 4a, The thermal~

In page 12: As shown in Figure 4xc, With~

In page 13, I don’t see the latent heat in Figure 5 (a,b)

In Page 14, Rewrite the second paragraph

In Page 15, there are errors in the first 3 line

In Page 16: comma looks weird

In Page 17: As shown in Figure 6

Add more detailed explanation how to measure thermal conductivity of CC4PCF-55. I guess that the composite is not isotropic, so the thermal conductivity strongly rely on the direction.

PEG is a PCM, which was encapsulated within the composite during composite fiber membrane. So the shape of membrane is sustained after phase change processes. Authors should clearly explain it. Additionally, the meaning of 'CC4PCF-55' needs clarification. It is the composite fiber membrane itself, not a PCM. It is not appropriate to characterize the entire composite material as a PCM. A more detailed explanation is necessary regarding how to measure the latent heat of the material.

Comments on the Quality of English Language

Included in the main comments

Author Response

Comments from the Reviewer 2:

This manuscript introduces a composite fiber membrane for PCM applications, which is a crucial topic in thermos engineering.

  1. While the topic is interesting, its value is diminished by the presence of unrefined English, numerous typos, and grammatical mistakes. For example,

In Page 3: can not (àcannot)

Introduction: When citing reference, write only family name of first author.

In page 8: Figure 2a-1

In Page 9: ~. 1g of ~ “In English writing, a sentence does not begin with number. (à One g of~)

In Page 11: “the change in the characteristic absorption peaks confirmed the acetal reaction was suceesfully conducted”

As shown in Figure 4a, The thermal~

In page 12: As shown in Figure 4c, With~

In page 13, I don’t see the latent heat in Figure 5 (a,b)

In Page 14, Rewrite the second paragraph

In Page 15, there are errors in the first 3 line

In Page 16: comma looks weird

In Page 17: As shown in Figure 6

Reply: We appreciate the reviewer's suggestion. We are very sorry for the many details due to our carelessness. In view of your reminder, we have modified the following:

the resulting material cannot be well adapted to the curve of the human skin

Hu et al. crosslinked styrene butadiene styrene …; Wu et al. prepared PEG/PVA/CNTs phase change films…; Zhang et al. prepared PEG/PVP/CNTs phase change fibers…

Figure 2 (a−m) showed that the composite fibrous membranes presented…

The composite fiber membrane of 1 g can successfully pull up to 500 g of weight.

The change in the characteristic absorption peaks confirmed the acetal reaction was successfully conducted.

As shown in Figure 4a, the thermal conductivity of PCF-55…

As shown in Figure 4c, with the passage of time,

Figure 5 (a, b) is the DSC curve, showing the latent heat absorption / release process during the phase transition of the sample.

As shown in Fig. 6c, the melt phase transition temperature of the composite fibre membranes decreased and their supercooling degree also reduced. This was due to the chemical cross-linking of PEG with PEO to promote the reduction of its melting phase transition temperature. And the addition of CNTs not only acted as a heterogeneous nucleating agent, but also as a heat transfer medium to improve the thermal conductivity and promote heat transfer [31]. The enthalpies of melting and crystallisation of C4PCF-55 are 69.37 J/g and 68.65 J/g, respectively; those of CC4PCF-55 are 66.71 J/g and 64.74 J/g. Compared with the calculated theoretical values (80.25 J/g and 78.32 J/g), they decreased by 13.6%, 12.3%, 16.9% and 17.3%, respectively. This suggested that a small portion of the PEG was not participated in the phase transition process during heating and freezing [32]. This was primarily attributed to the hydrogen bonding effect and acetal reaction between PEG and PEO, which resulted in a small portion of PEG chains existing in an amorphous state. Moreover, the solvent evaporated rapidly during centrifugal electrospinning and the PEG chains could not form a well-arranged crystal in the 3D polymer network. The cross-linking of GA with PEO and PEG further limited the movement of PEG and hindered the formation of crystalline reaction, which caused a further reduction of the enthalpy.

Figure 6. Thermal properties of the prepared composite fibre membranes. DSC curves of PEG and the composite fibre membranes for the endothermic process (a) and the exothermic process (b). (c) Melting/freezing enthalpies (ΔHm/ΔHf) and melting/freezing phase transition temperatures (Tm/Tf) of the composite fibre membranes. DSC curves of the CC4PCF-55 fibre membrane after 2, 20 and 50 cycles in the endothermic process (d) and exothermic process (e).

The organic substances as solvents are evaporated during the spinning process and emitted directly into the air, such as DMF, DMAC, formic acid, and acetone.

As shown in Figure 6, during the heating process,

  1. Add more detailed explanation how to measure thermal conductivity of CC4PCF-55. I guess that the composite is not isotropic, so the thermal conductivity strongly rely on the direction.

Reply: We appreciate the reviewer's suggestion. The thermal conductivity of all the samples in this work was determined by using a laser thermal conductivity meter. The laser thermal conductivity meter emits light pulses that are uniformly irradiated on the surface of the composite fibre membrane with an instantaneous increase in temperature. It acts as a hot end to transport the energy towards the cold end (upper surface) by one-dimensional heat conduction. An infrared thermometer then records the heating process on the upper surface to measure the heat diffusion coefficient and finally the thermal conductivity is calculated by combining the specific heat capacity and density of the composite fibre membrane. In practice, the composite fibre membrane is in contact with the device or human body on one side and the environment on the other. Therefore, the thermal conductivity of the composite fibre membrane measured by this method is more in conformity with the direction of thermal conductivity in reality.

  1. PEG is a PCM, which was encapsulated within the composite during composite fiber membrane. So the shape of membrane is sustained after phase change processes. Authors should clearly explain it. Additionally, the meaning of 'CC4PCF-55' needs clarification. It is the composite fiber membrane itself, not a PCM. It is not appropriate to characterize the entire composite material as a PCM. A more detailed explanation is necessary regarding how to measure the latent heat of the material.

Reply: We appreciate the reviewer's suggestion. In this study, a composite fibre membrane was prepared by centrifugal electrospinning of PEG and PEO after blending and cross-linking with each other. The prepared composite fibrous membrane possesses a stable structure due to the cross-linking of PEG and PEO, which transforms into a solid-solid phase transition and maintains a well-shaped stability without leakage during the phase transition. CC4PCF-55 is a composite fibre membrane containing PEG PCMs, which we have modified in the paper. The PEG PCMs and PEO in the composite fibre membrane are cross-linked and uniformly coexist with each other, so the composite fibre membrane was randomly selected for DSC experiments to characterise its latent heat. The specific testing conditions have been described in detail in Section 2.4.